# Unlocking Fairness: a Trade-off Revisited

**Michael Wick, Swetasudha Panda, Jean-Baptiste Tristan**
{michael.wick,swetasudha.panda,jean.baptiste.tristan}@oracle.com
Oracle Labs, Burlington, MA.

## Abstract

The prevailing wisdom is that a model's fairness and its accuracy are in tension with one another. However, there is a pernicious *modeling-evaluating dualism* bedeviling fair machine learning in which phenomena such as label bias are appropriately acknowledged as a source of unfairness when designing fair models, only to be tacitly abandoned when evaluating them. We investigate fairness and accuracy, but this time under a variety of controlled conditions in which we vary the amount and type of bias. We find, under reasonable assumptions, that the tension between fairness and accuracy is illusive, and vanishes as soon as we account for these phenomena during evaluation. Moreover, our results are consistent with an opposing conclusion: fairness and accuracy sometimes in accord. This raises the question, *might there be a way to harness fairness to improve accuracy after all?* Since many notions of fairness are with respect to the model's predictions and not the ground truth labels, this provides an opportunity to see if we can improve accuracy by harnessing appropriate notions of fairness over large quantities of *unlabeled* data with techniques like posterior regularization and generalized expectation. We find that semi-supervision improves both accuracy and fairness while imparting beneficial properties of the unlabeled data on the classifier.

## 1 Introduction

Torrents of ink have been spilled characterizing the relationship between a classifier's "fairness" and its accuracy [11, 7, 3, 8, 20, 4, 14, 2, 13, 17], where fairness refers to a concrete mathematical embodiment of some rule provided by an external party such as a government and which must be imposed on a learning algorithm. The consensus, countenanced by both empirical and analytical studies, is that the relationship is a trade-off: satisfying the supplied fairness constraints is achieved only at the expense of accuracy. On the one hand, these findings are intuitive: if we think of fairness as constraints limiting the set of possible classification assignments to those that are collectively fair, then clearly accuracy suffers because in general, optimization over the subset always lower bounds optimization over the original set. As put in another paper "demanding fairness of models *always* come at a cost of reduced accuracy" [2].[1]

On the other hand, the belief in a simple assumption immediately calls these findings into question. In particular, it requires no stretch of credulity to imagine that various personal attributes (e.g., race, gender, religion; sometimes termed "protected attributes") have no bearing on a person's intelligence, capability, potential, qualifications, etc., and consequently no bearing on ground truth classification labels — such as job qualification status — that might be functions of these qualities.[2] It then follows that enforcing fairness across these attributes should on average *increase* accuracy. The reason is clear. If our classifier produces different label distributions depending on the values of these dimensions, then we know, under the foregoing assumption, that at least one of these distributions must be wrong, and thus there is an opportunity to improve accuracy. An opportunity to which we later return.

But first we must understand what accounts for this antinomy. Two possible explanations involve the phenomena of label bias and selection bias. Label bias occurs when the process that produces the labels (e.g., a manual annotation process or a decision making process such as hiring) are influenced by factors that are not particularly germane to the determination of the label value, and thus might differ from the ideal labels, whatever they should have been. Accuracy measured against any such biased labels should be considered carefully with a grain of salt. Selection bias occurs when selecting a subsample of the data in such a way that happens to introduce unexpected correlations, say, between a protected attribute and the target label. Training data, which is usually derived via selection from a larger set of unlabeled data and subsequently frozen in time, is especially prone to this problem.

If pressed to couch the above discussion in a formal framework such as probably approximately correct (PAC) learning, we would say that we have a data distribution $\mathcal{D}$ and labeling function $f$, either of which could be biased. For example, due to selection bias we might have a flawed data distribution $\mathcal{D}'$ and due to label bias we might have a flawed labeling function $f'$. This leads to four regimes: the data distribution is biased ($\mathcal{D}'$) or not ($\mathcal{D}$) and the labeling function is biased ($f'$) or not ($f$). Many theoretical works in fair machine learning consider the regime in which neither is biased, and many empirical works—due in part to the unavailability of an unbiased $f$—draw conclusions assuming the regime in which neither is biased. But many forms of unfairness arise exactly because one or both of these are biased: hence the dualism in fair machine learning. In this work, we assume that some of the unfairness might arrise because we are actually in one of the other three regimes.

In this paper we account for both label and selection bias in our evaluations and show that when taken into consideration, that certain definitions of fairness and accuracy are not always in tension. Since we do not have access to the unbiased, unobserved ground truth labels in practice, we instead simulate datasets in tightly controlled ways such that, for example, it exposes the actual unbiased labels for evaluation. Encouraged by theoretical results on semi-supervised PAC learning that state that these techniques will be successful exactly when there is compatibility between some semi-supervised signal and the data distribution [1] and the success of GE [16, 10], we also introduce and study a semi-supervised method that exploits fairness constraints expressed over large quantities of unlabeled data to build better classifiers. Indeed, we find that as fairness improves, so does accuracy. Moreover, we find that like other fairness methods, the semi-supervised approach can successfully overcome label bias; but unlike other fairness methods, it can also overcome selection bias on the training set.

## 2  Related work

Somehow, the idea that fairness and accuracy are not always in tension is both obvious and inconspicuous (but nevertheless of practical significance). The idea appears obvious because we assume the unobserved unbiased ground-truth to be fair, and then limit our hypotheses to the fair region of the space, and then claim that fairness improves accuracy. At this level of generality, it even appears to beg the question, but note that not all fair hypotheses are accurate since in the case we consider a perfectly random classifier is also fair. Moreover, the noise on the observed biased labels with which we train the classifier is diametrically opposed to the unobserved label. Thus even under our assumptions, it is not a foregone conclusion that improving fairness improves accuracy. Rather, our assumption merely leaves open the possibility for this to happen. The finding is inconspicuous in the sense that, as mentioned earlier, there is a preponderance of work investigating this trade-off yet label bias appears to have gone unnoticed: very few papers (e.g., [8, 20]) mention the fact that the labels against which we evaluate are often biased (unfairly against a protected attribute) in the very same way as the unfair classifier trained on them [11, 7, 3, 8, 20, 4, 14, 2, 13, 17]. It may be the case that label-bias is so obvious to most authors that it does not even occur to them to mention it; howbeit, the conspicuous absence of label-bias from papers on fairness perniciously pervades real-world discussions underlying the decisions about how to balance the trade-off between fairness and accuracy. Thus, we believe this finding to be of practical importance and worthy of highlighting.

While uncommon, some papers do indeed mention label-bias, including recent work that considers the largely hypothetical case: if we have access to unbiased labels, then we can propose a better way of evaluating fairness with "disparate mistreatment" [20]. However, their emphasis is on new fairness metrics, not on its tradeoff with accuracy. Other work mentions the problem of label bias in passing, lamenting that it is difficult to account for in practice because we "only have biased data" and thus we "cannot evaluate our classifiers against an unbiased ground truth" and so achieving parity requires

that "one must be willing to reduce accuracy" ([8]). They overcome the lack of unbiased labels via data simulation, a strategy we also employ.

Congruent with our findings, others have noted that the fairness-accuracy tension is not as bad as it seems. Recent work correctly remarks that while there is a tradeoff between fairness and goodness of fit on the training set, that "it does not [necessarily] introduce a tension" since a reduction in model complexity via fairness constraints might act as a regularizer and improve generalization [2]. This is a very interesting remark, but it could have gone even further and addressed generalization with respect to the unbiased labels, which we study in this work.

In recent theoretical work, the authors' propose a "construct space" in which the observed data might differ from some unobserved actual truth about the world [9]. While they investigate many different notions of fairness, they do not address accuracy. The construct space provides a promising theoretical framework for our work, but we save such analysis for another day. Other analytical work studies the trade-off between fairness and accuracy as a function of the amount of statistical dependence between the target class and protected attribute, concluding that only "in the other extreme" of perfect independence that "we can have maximum accuracy and fairness simultaneously" [17]. This "extreme" is none other than the "we're all equal" assumption, which we believe to be perfectly reasonable in many situations. Further, note that this theoretical "maximum" may not be achievable in practice due to imperfect classifiers trained on incomplete, noisy data, or in the context of the phenomena mentioned herein, and hence there is still an opportunity to improve both.

It is worth thinking about the problems of selection and label bias with respect to an existing fairness datasets such as COMPAS, for which the labels are sometimes treated as if they are the unbiased ground truth [20]. Consider that the people in the COMPAS data had been selected from a specific county in Florida with its concomitant pattern of policing, during a specific period of time (2013-2014), meeting a specific set of criteria such as being scored during a specific stage within the judicial system. Each one of these "selections" opens the door for selection bias to introduce unintentional correlations. Indeed, recent work demonstrates that the data is skewed with respect to age, which acts as a confounding variable in existing analysis [18]. Moreover, while not exactly label-bias, the variable indicating recidivism is only partially observed since it considers only a two-year window and assumes that no crime goes uncaught.

Finally, we emphasize that our findings do not imply that the existing theories and conclusions discussed in the literature are incorrect. On the contrary, these works are in fact both sound and relevant. The different conclusions then are explained by the consideration of different types of data bias (or lack thereof) as well as the underlying assumptions, and our assumptions may not always apply [3]. If there differences between groups based on a protected attribute (e.g., due to selection bias), then enforcing fairness could indeed hurt accuracy. We do not address the degree to which one assumption applies to a particular problem or dataset in this paper. Thus, just like in statistical significance testing, it remains up to the discretion of the discerning practitioner to determine if our (or their) set assumptions reasonably apply to the situation in question, and if the assumptions do not, then our (or their) conclusions do not apply, and should be properly rejected as irrelevant to that data.

## 3 Background

**Fairness and bias types**   We consider two types of biases that lead to unfair machine learning models: label bias and selection bias. Label bias is when the observed binary class labels, say, on the training and testing set, are influenced by protected attributes. For example, the labels in the dataset might be the result of yes/no hiring decisions for each job candidate. It is known that this hiring process is sometimes biased with respect to protected attributes such as race, age or gender. Since decisions might be influenced by protected attributes that on the contrary should have no bearing on the class label, this implies there is a hypothetical set of latent unobserved labels corresponding to decisions that were not influenced by these attributes. We notate these unobserved unbiased labels as $z$. We notate the observed biased labels as $y$. Typically, we only have access to the latter for training and testing our models.

Selection bias (skew) occurs when the method employed to select some subset of the overall population biases or skews the subset in unexpected ways. This can occur if selecting based on some attribute that inadvertently correlates with a protected class or the target labels. Training sets are particularly vulnerable to such bias because, for the sake of manual labeling expedience, they are

meager subsamples of the original unlabeled data points. Moreover, this problem is compounded since most available labeled datasets are statically frozen in time and are thus also selectionally biased along the axis of time. For example, in natural language processing (NLP), the particular topical subjects or the entities mentioned in newswire articles change over time: the entities discussed in political discourse today are very different from a decade ago and new topics must emerge to keep pace with the dernier cri [19]. And, as we continue to make progress in reducing discrimination, the discrepancy between the training data of the past and the available data of the present will increasingly differ w.r.t. to selection bias. Indeed, selection bias might manifest itself in a way such that on the relatively small training set, the data examples that were selected for labeling happen to show bias against the protected class. It is with this manifestation of selection bias that we are most concerned, and that we study in the current work.

**Illustrative example: learning fair sectors**  Consider the problem of learning circular sectors of the unit disk with the following attributes: the domain set $\mathcal{X}$ is the unit disk, the label set $\mathcal{Y}$ is $\{0,1\}$, the data generation model $\mathcal{D}$ is an arbitrary density on $\mathcal{X}$, the labeling function $f$ is an arbitrary partition of $\mathcal{X}$ into two circular sectors, the hypothesis class $\mathcal{H}$ is the set of all partitions of $\mathcal{X}$ into two circular sectors. Samples from $\mathcal{D}$ are points on the unit disk with location $re^{i\phi}$ where $\phi \in [0, 360)$ and $r \in [0,1]$. We represent a circular sector as a pair of angles $(\mu, \theta)$ and defined as the circular sector from angle $(\mu - \theta)\%360$ to angle $(\mu + \theta)\%360$ that contains the point $e^{i\mu}$. The labeling function $f$ partitions the disk in two circular sectors $f^{-1}(0)$ and $f^{-1}(1)$ and we will refer to the former as the negative circular sector and the latter as the positive circular sector. Note that for any labeling function $f$, we have $f \in \mathcal{H}$ and so the realizable assumption holds.

Due to label bias, the labeling function $f$ might be biased ($f'$) as shown in Figure 1. Here, the total positive area according to $f$ is given by the area in green and red, but because of label bias $f'$ only considers points in green as positive. Hence, as demonstrated in Figure 2, an empirical risk minimization (ERM) algorithm will learn a sector (dotted lines) that appears accurate with respect to $f'$, but is much less accurate with respect to $f$. If we had prior knowledge that the ratio of the positive sector and negative sector should be some constant $k$, perhaps we could exploit this and improve the ERM solution. We might term such an alternative empirical fairness maximization (EFM) (or fair ERM [5]), and in this paper, we present a semi-supervised EFM algorithm to exploit such fairness knowledge as a constraint on unlabeled data. This example is fully developed in appendix B.

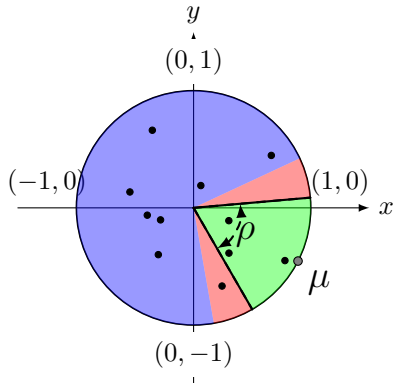

Figure 1

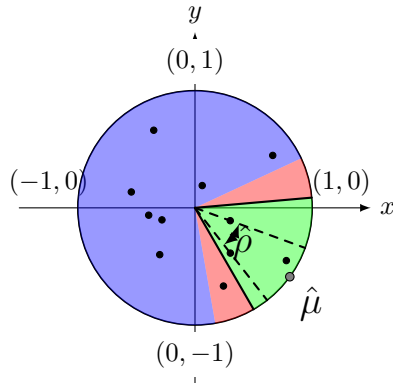

Figure 2

**Semi-supervised classification**  A binary classifier[3] $g_w : \mathbb{R}^k \to \{0,1\}$ parameterized by a set of weights $w \in \mathbb{R}^k$ is a function from a $k$ dimensional real valued feature space, which is often in practice binary, to a binary class label. A probabilistic model $p_w(\cdot|x)$ parameterized by (the very same) $w$ underlies the classifier in the sense that we perform classification by selecting the class label (0 or 1) that maximizes the conditional probability of the label $y$ given the data point $x$

$$g_w(x) = \underset{y \in \{0,1\}}{\operatorname{argmax}} \, p_w(y|x) \tag{1}$$

We can then train the classifier in the usual supervised manner by training the underlying model to assign high probability to each observed label $y_i$ in the training data $\mathcal{D}_{\text{tr}} = \{\langle x_i, y_i \rangle \mid i = 1 \ldots n\}$ given the corresponding example $x_i$, by minimizing the negative log likelihood:

$$\hat{w} = \operatorname*{argmin}_{w \in \mathbb{R}^k} \sum_{\langle x_i, y_i \rangle \in \mathcal{D}_{\text{tr}}} - \log p_w(y_i | x_i) \tag{2}$$

We can extend the above objective function to include unlabeled data $\mathcal{D}_{\text{un}} = \{x_i\}_{i=1}^n$ to make the classifier semi-supervised. In particular, we add a new term to the loss, $\mathcal{C}(\mathcal{D}_{\text{un}}, w)$, with a weight $\eta$ to control the influence of the unlabeled data over the learned weights:

$$\hat{w} = \operatorname*{argmin}_{w \in \mathbb{R}^k} \left( \sum_{\langle x_i, y_i \rangle \in \mathcal{D}_{\text{tr}}} - \log p_w(y_i | x_i) \right) + \eta \mathcal{C}(\mathcal{D}_{\text{un}}, w) \tag{3}$$

The key question is how to define the loss term $\mathcal{C}$ over the unlabeled data in such a way that improves over our classifier.

## 4 Approach

Apropos the foregoing discussion, we propose to employ fairness in the part of the loss function that exploits the unlabeled data. There are of course many definitions of fairness proposed in the literature that we could adapt for this purpose, but for now we focus on a particular type of group fairness constraint derived from the *statistical parity* of selection rates. Although this definition has (rightfully) been criticized, it has also (rightfully) been advocated in the literature and it underlies legal definitions such as the 4/5ths rule in U.S. law [6, 8, 21]. For the purpose of this paper, we do not wish to enter the fray on this particular matter.

More formally, let $S = \{x_i\}_{i=1}^n$ be a set of $n$ unlabeled examples, then the selection rate of the classifier $g_w$ is $\bar{g}_w(S) = \frac{1}{n} \sum_{x_i \in S} g_w(x_i)$. If we partition our data ($\mathcal{D}_{\text{un}}$) into the protected ($\mathcal{D}_{\text{un}}^P$) and unprotected ($\mathcal{D}_{\text{un}}^U$) partitions such that $\mathcal{D}_{\text{un}} = \mathcal{D}_{\text{un}}^P \cup \mathcal{D}_{\text{un}}^U$, then we want the selection rate ratio

$$\frac{\bar{g}_w(\mathcal{D}_{\text{un}}^P)}{\bar{g}_w(\mathcal{D}_{\text{un}}^U)} \tag{4}$$

to be as close to one as possible. However, to make the problem more amenable to optimization via stochastic gradient descent, we relax this definition of fairness to make it differentiable with respect to $w$. In particular, analogous to $\bar{g}_w(S)$, define $\bar{p}_w(S) = \frac{1}{n} \sum_{x_i \in S} p_w(y = 1 | x_i)$ to be the average probability of the set when assigning each example $x_i$ to the positive class $y_i = 1$. Then, the group fairness loss over the unlabeled data — which when plugged into Equation 3 yields an instantiation of the proposed semisupervised training technique discussed herein — is

$$\mathcal{C}(\mathcal{D}_{\text{un}}, w) = \left( \bar{p}_w(\mathcal{D}_{\text{un}}^P) - \bar{p}_w(\mathcal{D}_{\text{un}}^U) \right)^2 \tag{5}$$

Parity is achieved at zero, which intuitively encodes that overall, the probability of assigning one group to the positive class should on average be the same as assigning the other group to the positive class. This loss has the important property that it is differentiable with respect to $w$ so we can optimize it with stochastic gradient descent, along with the supervised term of the objective, making it easy to implement in existing toolkits such as Scikit-Learn, PyTorch or TensorFlow.

## 5 Experiments

In this section we investigate the relationship between fairness and accuracy under conditions in which we can account for (and vary) the amount of label bias, selection bias, and the extent to which the classifiers enforce fairness. Typically, accuracy is measured against the ground truth labels on the test set, which inconspicuously possesses the very same label bias as the training set. In this typical evaluation setting, if we train a set of classifiers that differ only in the extent to which their training objective functions enforce fairness, and then record their respective fairness and accuracy scores on a test set with such label bias, we see that increased fairness is achieved at the expense

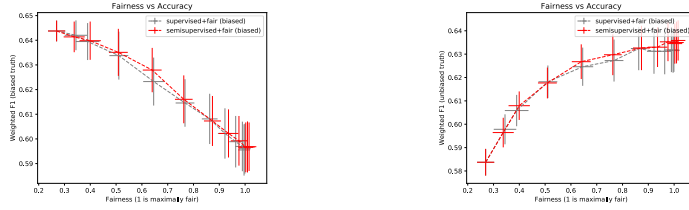

(a) COMPAS (biased ground truth)   (b) COMPAS (unbiased ground truth)

Figure 3: Accuracy vs. fairness on simulated ($\beta$=0.25) COMPAS (assumption hold).

of accuracy (Figure 3a). However, because the labels are biased, we must immediately assume that the corresponding accuracy measurements are also biased. Therefore, we are crucially interested in evaluating accuracy on the *unbiased ground truth labels*, which are devoid of any such label bias. Since we do not have access to the unbiased ground truth labels of real-world datasets, we must instead rely upon data simulation. We discuss the details later, but for now, assume we could evaluate on such data. In Figure 3a, we evaluate the same set of classifiers as before, but this time measure accuracy with respect to the unbiased ground truth labels. We see the exact opposite pattern: classifiers that are more fair are also more accurate. With the gist of our results and experimental strategy in hand, we are now ready to describe the assumptions, data simulator, and systems to undertake a more comprehensive empirical investigation.

**Assumptions**   We make a set of assumptions that we encode directly into the probabilistic data generator, explained in more detail below. For example, we encode the "we're all equal assumption" by making the unbiased labels statistically independent of the protected class [9]. If these assumptions do not hold in a particular situation, then our conclusions may not apply. We describe the assumptions in more detail below and in the appendix.

**Data**   Our experiments require datasets with points of the form $\mathcal{D} = \{x, \rho, z, y\}$ in which $x$ is the vector of unprotected attributes, $\rho$ is the binary protected attribute, $z$ is the (typically unobserved) label that has no label bias and $y$ is the (typically observed) label that may have label bias. Since $z$ is unobserved — and even if it were available, we would still want to vary the severity of label bias for experimental evaluation — we must rely upon data simulation [8]. We therefore assume that the biased labels are generated from the unbiased labels via a probabilistic model $g$ and assume that $y \sim g(y|z, \rho, x, \beta)$ where $\beta$ is a parameter of the model that controls the probability of label bias occurring. Now we have two options for generating datasets of our desired form, we can either (a) simulate the dataset entirely from scratch from a probabilistic model of the joint distribution $P(x, \rho, z, y) = g(y|z, \rho, x, \beta)P(z, \rho, x)P(\beta)$, or we can (b) begin with an existing dataset, declare by fiat that the labels have no label bias (and are thus observed after all) and then augment the data with a set of biased labels sampled from $g(y|z, \rho, x, \beta)$.

For data of type (a) we generate the features and labels (both biased and unbiased) entirely from scratch with the Bayesian network in Figure 7 (Appendix A.2). For this data, we explicitly enforce the following statistical assumptions: $z, x \perp\!\!\!\perp \rho, y \not\perp\!\!\!\perp \rho, z \not\perp\!\!\!\perp x, y \not\perp\!\!\!\perp z$. A parameter $\beta$ controls the amount of label bias; $\sigma$ controls the amount of selection bias, which can break some assumptions. For data of type (b) we begin with the COMPAS data, treat the two-year recidivism labels as the unbiased ground-truth $z$ and then apply our model of label bias to produce the biased labels $y \sim g(z|y, \rho, x, \beta)$ [15]. Since the "we're all equal" assumption does not hold for COMPAS data we also create a second type of test data in which we enforce demographic parity via subsampling so that our assumption holds (see Appendix A.3).

**Systems, baselines and evaluations**   We study the behavior of the following classification systems. A traditional supervised classifier trained on biased label data, a supervised classifier trained on unbiased label data (this in some sense is an ideal model, but not possible in practice because we do not have access to the unbiased labels in practice), a random baseline in which labels are sampled according to the biased label distribution in the training data, and three fair classifiers. The first fair classification method is an in-processing classifier that employs our fairness constraint, but as a regularizer on the training data instead of the unlabeled data. The resulting classifier is similar

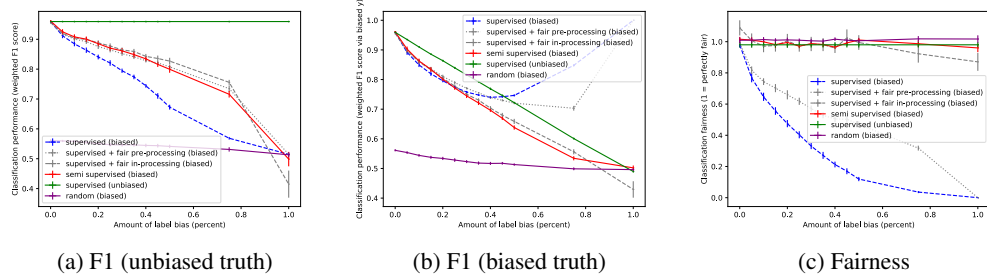

(a) F1 (unbiased truth)   (b) F1 (biased truth)   (c) Fairness

Figure 4: Classifier accuracy (F1) and fairness as a function of the amount of label bias.

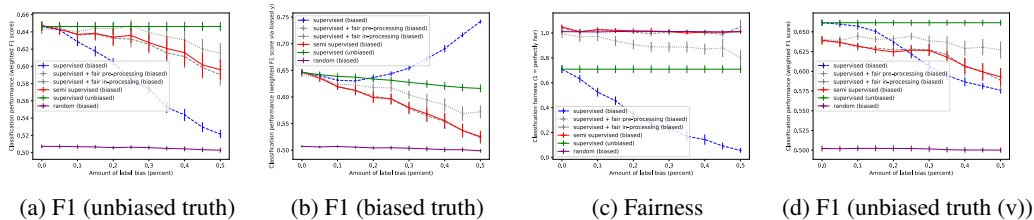

(a) F1 (unbiased truth)   (b) F1 (biased truth)   (c) Fairness   (d) F1 (unbiased truth (v))

Figure 5: Varying label bias on COMPAS (assumption holds, except in 5d).

to the prejudice remover, but with a slightly different loss [12]. The second fair classifier is a supervised logistic regression trained using the "reweighing" pre-processing method [11]. The final fair classifier, which we introduce in this paper, is a semi-supervised classifier that utilizes the fairness loss (Equation 5) on the unlabeled data.

We assess fairness with a group metric that computes the ratio of the selection rates of the protected and unprotected class, as we defined in Equation 4. A score of one is considered perfectly fair. To assess 'accuracy' we compute the weighted macro F1, which is the macro average weighted by the relative portion of examples belonging to the positive and negative classes. We evaluate F1 with respect to both the biased labels and the unbiased labels. We always report the mean and standard error of these various metrics computed over ten experiments with ten randomly generated datasets (or in the case of COMPAS, ten random splits).

## 5.1 Experiment 1: Label Bias

In this experiment we investigate the relationship between fairness and accuracy for each classification method as we vary the amount of label bias. All classifiers except the unbiased baseline are trained on biased labels. If we evaluate the classifiers on the biased labels as in Figure 4b (data simulated from scratch) or Figure 5b (COMPAS data) we see that the classifiers that achieve high fairness (close to one, as seen in Figure 4c&5c) sometimes degrade the (biased) F1 accuracy as commonly seen in the literature. On the other hand, if we evaluate the classifiers on the unbiased labels as in Figure 4a&5a, we see that fairness and accuracy are in accord: the classifiers that achieve high fairness achieve better accuracy than the fairness-agnostic supervised baseline. The gap between the fair and unfair classifiers increases as label bias increases. We also evaluate the classifiers on COMPAS data that violates the "we're all equal" assumption. In this case, the fairness classifiers are enforcing something untrue about the data, and thus fairness initially degrades accuracy (Figure 5d). However, as the amount of label bias increases, eventually there comes a point at which fairness once again improves accuracy (possibly because the amount of label bias exceeds the amount of other forms of bias).

## 5.2 Experiment 2: Selection Bias

We repeat the experiment from the last section, but this time fixing label bias ($\beta = 0.2$) and subjecting the training data to various amounts of selection bias by lowering the probability that a data example with a positive label is assigned to the protected class. This introduces correlations in the training set between the protected class and the input features as well as correlations with both the unbiased and

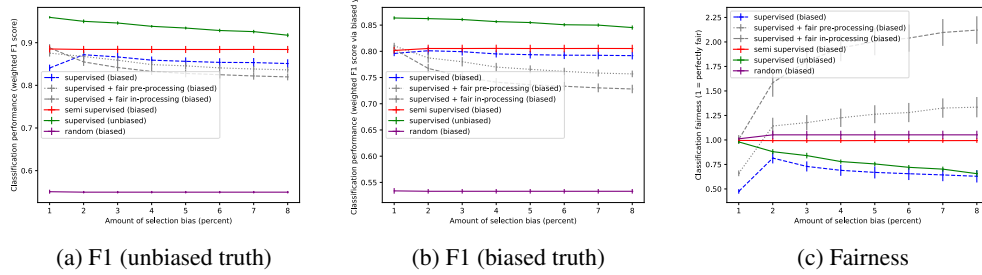

|                     |                    |                 |
|:-------------------:|:------------------:|:---------------:|
| (a) F1 (unbiased truth) | (b) F1 (biased truth) | (c) Fairness |

Figure 6: Classifier accuracy (F1) and fairness as a function of the amount of selection bias.

biased labels. These correlations do not exist in the test set and unlabeled set which we assume do not suffer from selection bias. We vary selection bias along the abscissa while keeping the label bias at a constant 20%, and report the same metrics as before. Results are in Figure 6. The main findings are that (a) the results are consistent with the theory that fairness and accuracy are in accord and (b) that the semi-supervised method succesfully harnesses unlabeled data to correct for the selection and label bias in the training data (while the inprocessing fairness method succumbs to the difference in data distribution between training and testing). Let us now look at these findings in more detail and in the context of the other baselines.

Interestingly, the fairness-agnostic classifiers and two of the fairness-aware classifiers (in- and pre-processing) all succumb to selection bias, but in opposite ways (Figure 6c). The fairness-agnostic classifier learns the correlation between the protected attribute and the label and is unfair to the protected class. In contrast, the two supervised fair classifiers, for which fairness is enforced with statistics of the *training set* both learn to overcompensate and are unfair to the unprotected class (its fairness curve is above 1). In both cases, as selection bias increases, so does unfairness and this results in a concomitant loss in accuracy (when evaluated not only against the unbiased labels (Figure 6a), *but also against the biased labels* (Figure 6b)), indicating that fairness and accuracy are in accord. Finally, let us direct our attention to the performance of the proposed semi-supervised method by examining the same figures (Figure 6c). Now we see that regardless of the amount of selection bias, the semi-supervised method successfully harnesses the unbiased unlabeled data to rectify it, as seen by the flat fairness curve achieving a nearly perfect 1 (Figure 6c). Moreover, this improvement in fairness over the supervised baseline (biased trained) is associated with a corresponding increase in accuracy relative to that same baseline (Figures 6a & 6b), regardless of whether it is evaluated with respect to biased (20% label-bias) or unbiased labels (0% label-bias). Note that the "we're all equal" assumption is violated as soon as we evaluate against the biased labels. Moreover, the label-bias induces a correlation between the protected class and the target label, which is a common assumption for analysis showing that fairness and accuracy are in tension [17]. Yet, the beneficial relationship between accuracy and fairness is unsullied by the incorrect assumption in this particular case.

## 6 Conclusion

We studied the relationship between fairness and accuracy while controlling for label and selection bias and found that under certain conditions the relationship is not a trade-off but rather one that is mutually beneficial: fairness and accuracy improve together. We focused on demographic parity in this paper, but the ideas emphasized in this work, especially label bias, have potentially serious implications for other notions of fairness that go beyond even their relationship with accuracy. In particular, recent ways of assessing fairness such as disparate mistreatment, equal odds and equal oppurtunity involve error rates as measured against labeled data. Label bias raises questions about the reliability of such measures and investigating such questions — about how label bias affects fairness and whether this causes fairness methods to undercompensate or overcompensate — is an important direction of future work. Other future directions would be to develop more complex models of label and selection bias for particular domains so we can better understand the relationship between fairness and accuracy in these domains.

## 7 Acknowledgements

We thank the anonymous reviewers for their constructive feedback and helpful suggestions on how to strengthen the paper.

## Footnotes

[1]Our emphasis.

[2]This assumption is consistent with the "we're all equal" worldview [9]

[3]For ease of explication, we consider the task of binary classification, though our method can easily be generalized to multiclass classification, multilabel classification, or more complex structured prediction settings.

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
