[Supplementary Material]

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

. For example, in one experiment we assume no selection bias; in another, we assume the training data experiences more selection bias than the test data. If the opposite were true then, perhaps, a post-processing method of enforcing fairness might be more appropriate. We also make a "we're all equal" assumption [9], which we encode by ensuring that the unbiased labels are statistically independent of the protected dimension. Again, if this assumption is violated to a sufficiently large degree, then our conclusions do not apply. This assumption is important, but not necessary for the finding that fairness and accuracy are not always in tension. Finally, our framework optimistically presupposes that it is possible to model the way in which these biases actually infiltrate real-world datasets. For the purpose of this initial study, we employ the simplest possible models of biases — perhaps at the risk of oversimplifying — that still strongly capture their baleful effects, which researchers in fair ML toil to address.

## A.2 Simulated Data (non-COMPAS)

Here we provide details of our data simulation process. In particular, we enforce that the unobserved unbiased labels $z_i$ do not statistically depend on the example's status as protected (or not) $\rho_i$ and only on its other input features $x_i$. Second, the protected status $\rho$ does not depend on any features $x_i$. Third, the observed biased labels $y_i$ are biased to depend on the protected status $\rho$ by an amount controlled by the experimental parameter $\beta$, which we vary in our experiments.

In summary, we enforce the following statistical properties:

$$z, x \perp\!\!\!\perp \rho \qquad\qquad y \not\!\perp\!\!\!\perp \rho$$
$$z \not\!\perp\!\!\!\perp x \qquad\qquad y \not\!\perp\!\!\!\perp z$$

Where $\perp\!\!\!\perp$ (respectively, $\not\!\perp\!\!\!\perp$) are the familiar symbols for expressing statistical independence (respectively, dependence) of random variables. Note when we later introduce selection bias, it will break some of these independence assumptions (between $\rho$ and $z, x$, but in a controlled manner, so we then show to what extent we correct this via unlabeled data, as we vary the amount of selection bias.

To simulate the dataset, we sample the input data points $x$ iid from a $k$ dimensional binary feature space. We sample such that some dimensions contain common features while others contain rare features, in effort to reflect that real-world datasets. In particular, we sample each dimension $i$ according to a Bernoulli proportional to $\frac{1}{i}$ making some dimensions common and others rare.

The parameters of our data generator are $\beta$ the amount of label bias, $\tau$, which controls the discrepancy between the rarity of features, and $\alpha$, which controls the ratio between members of the protected and unprotected class. We also introduce a parameter $\sigma$ that controls the amount of selection bias. First, we sample the observed samples $x_i$ and its status as protected ($\rho = 1$) or not ($\rho = 0$), independently to ensure that protected status and input features are not statistically dependent. Next, we sample the unobserved unbiased labels $z$ from $x_i$ while crucially ignoring the protected status $\rho_i$ to ensure that the label is indeed unbiased. Finally, we sample the observed biased labels $y$ in a way to make them dependent on the class labels $\rho_i$, a dependency strength controlled by $\beta$. More precisely:

$$w_{\text{gen}} \sim N(\mathbf{0}, \Sigma) \tag{6}$$
$$\rho_i \sim \text{Bernoulli}(\alpha) \tag{7}$$
$$x_i^j \sim \text{Bernoulli}\left(\frac{1}{j+1}\right)^{\tau} \text{ for } j = 0, \dots, k-2 \tag{8}$$
$$z_i = \max(0, \text{sign}\left(w_{\text{gen}}^T x_i\right)) \tag{9}$$
$$y_i \sim g(y|z_i, \rho_i, x_i, \beta) \tag{10}$$

Figure 7: Data generator as a Bayesian network.

where $g$ is the label bias model, parameterized by $\beta \in [0, 1]$, the amount of label bias to introduce, and is a function of the protected dimension and the unobserved unbiased labels $z_i$, and defined as

$$g(y_i|z_i, \rho_i, x_i, \beta) = g(y_i|z_i, \rho_i, \beta) = \begin{cases} \beta & \text{if} \quad y_i \neq z_i \wedge z = \rho_i \\ 1 - \beta & \text{o.w.} \end{cases} \tag{11}$$

This model assumes that the desirable label is 1 (say calling a candidate for an interview, or offering a loan) and that the bias will be against the protected class and in favor of the unprotected class. Hence with probability $\beta$, a protected class individual that has an unbiased label of 1 will have it flipped to 0; similarly, an unprotected class individual that has an unbiased label of 0 will have it flipped favorably to one with probability $\beta$. Note the model is simplistic in that it does not make use of the unprotected features and that it assumes symmetry in the bias, as just described. Note that other models could be used for specific datasets or problem domains for which a domain expert has a theory or insight about what the nature of the label bias might be.

The function returns the unbiased labels with probability $1 - \beta$, but otherwise works *against* examples of the protected class by assigning their labels to 0, and *for* all other examples by assigning their labels to 1. See Figure 7 for a Bayesian network representation of the generator.

We can control the amount of selection bias with a parameter $\sigma$. As selection bias increases, the data is increasingly unlikely to contain members of the protected attribute ($\rho = 1$) that have a favorable class label ($z = 1$). In particular, if $r \in [0, 1]$ is the portion of the protected attribute assigned to the favorable class in the original data, then the selection bias process "selects" the data in such a way to reduce this portion to $\frac{r}{\sigma}$ for $\sigma \geq 1$. Therefore if $\sigma = 1$ then no bias occurs, and if $\sigma > 1$ then an amount of bias against the protected class occurs proportional to $\sigma$. Note that this selection procedure introduces statistical dependencies between the input $x$ and the unbiased label $z$ as well as between the protected class $\rho$ and the unbiased label.

For the non-COMPAS experiments, we generate 20-dimensional binary input features $x_i$ and, 200 training examples, 1000 testing examples and 10,000 unlabeled examples. Note that 200 training examples is reasonable since it means that $n \gg k$ as is usually required for training machine learning models. Yet, at the same time, it is small enough to allow for the weights of some of the rarer features to be under-fit as is typical in most applications of machine learning. Also, unless otherwise stated, the expected protected to unprotected class ratio is even at 50/50, though we have repeated the experiments but with a skewed expected ratio of 20/80 and found it did not affect the conclusions. We train each classifier with 10 epochs of stochastic gradient descent, which we found to be sufficient for this dataset.

### A.3 COMPAS Data with Simulated Bias

The COMPAS data is a dataset of criminal recidivism. Here, the task is to predict recidivism (after two-years) from a set of demographic features including age (under 25, over 45 and between 25 and

45), sex, race, prior count (0-37), charge degree (misconduct or felony). We employ race (African-American or not) as the binary protected attribute. A key challenge with real-world data such as COMPAS is that it exhibits both selection and label bias, thus making it difficult to perform our evaluations, which crucially rely on the existence of an *unbiased test set*. To this end, we resort to a combination of simulation and subsampling to unbias the test set.

First, we assume that there is no label bias in the two-year recidivism labels, but then create a biased version of the labels in a simular fashion as before (in this case, by randomly flipping the recidivism label for African Americans from 0 to 1, and by randomly flipping the recidivism label for all others from 1 to 0). In this way, we can create a discrepancy in label bias between the training and testing data. Second, we can force the test set to adhere to the "we're all equal" assumption by subsampling the data such that the recidivism rates are the same for the protected and unprotected class. We thus have two versions of the test data, one in which we enforce this assumption and one in which we do not. In either case, we can vary the amount of label bias on the training set in the same way.

For our COMPAS experiments, we perform ten random splits of the data (7215 total examples) in which we partition the data into 40% train 40% unlabeled and 20% test. Each algorithm is fit with stochastic gradient descent, trained for two epochs on the training data. We found that two epochs was sufficient for training, likely because the training set is large (almost 3000 examples) relative to its dimensionality (about 50 features).

# B    Learning Circular Sectors: Complete Example

We present the completely developed example mentioned in the paper on learning circular sector with a potentially biased labeling function. This illustrates precisely the intuitive idea that when the labeling function is biased, then ERM's guarantees with respect to the true labeling function are void, while trying to maximize fairness has very strong guarantees.

## B.1   Learning Circular Sectors

We consider the problem of learning circular sectors of the unit disk with the following attributes. The domain set $\mathcal{X}$ is the unit disk, the label set $\mathcal{Y}$ is $\{0, 1\}$, the data generation model $\mathcal{D}$ is an arbitrary density on $\mathcal{X}$, the labeling function $f$ is an arbitrary partition of $\mathcal{X}$ into two circular sectors, the hypothesis class $\mathcal{H}$ is the set of all partitions of $\mathcal{X}$ into two circular sectors.

Samples from $\mathcal{D}$ are points on the unit disk with location $re^{i\phi}$ where $\phi \in [0, 360)$ and $r \in [0, 1]$. We represent a circular sector as a pair of angles $(\mu, \theta)$ and defined as the circular sector from angle $(\mu - \theta)\%360$ to angle $(\mu + \theta)\%360$ that contains the point $e^{i\mu}$. The labeling function $f$ partitions the disk in two circular sectors $f^{-1}(0)$ and $f^{-1}(1)$ and we will refer to the former as the negative circular sector and the later as the positive circular sector. Note that for any labeling function $f$, we have $f \in \mathcal{H}$ and so the realizable assumption holds.

Figure 8: Domain set $\mathcal{X}$ with the positive region of location $\mu$ and spread $\theta$ in green and the negative region in blue.

Figure 9: Samples from $\mathcal{D}$ labeled by $f$.

```
input        : A set S of N labeled examples (r_j e^{iφ_j}, l_j) for j ∈ [N]
precondition : S contains at least 3 positive examples with distinct angles
output       : A hypothesis h ∈ H
1  a = max{φ_j | (r_j e^{iφ_j}, l_j) ∈ S ∧ l_j = 1};
2  b = min{φ_j | (r_j e^{iφ_j}, l_j) ∈ S ∧ l_j = 1};
3  c = choose{φ_j | (r_j e^{iφ_j}, l_j) ∈ S ∧ φ_j ≠ a ∧ φ_j ≠ b};
4  if b < c < a then
5  |    θ = a − b;
6  |    μ = b + θ/2;
7  else
8  |    θ = 360 − a + b;
9  |    μ = a + θ/2;
10 end
11 return (e^{iμ}, θ)
```

**Algorithm 1:** $A_{ERM}$, Smallest positive circular sector

The smallest positive circular sector algorithm implements empirical risk minimization and is probably approximately correct.

Figure 10: Error of $A_{ERM}$

**Theorem 1** (The $A_{ERM}$ algorithm is probably approximately correct). *For any accuracy $\epsilon > 0$ and confidence $0 < \delta < 1$, there exists a finite number $m$ such that if we take $m$ independent samples*

$e_1, ..., e_m$ *from* $\mathcal{D}$ *and let* $h = A_{ERM}(\{(e_i, f(e_i))\})$ *we have*

$$\Pr\left[\mathcal{D}(\{x \mid h(x) \neq f(x)\}) > \epsilon\right] \leq 1 - \delta$$

*Proof.* In the following proof, we assume that we have at least 3 examples with distinct angles, which is a fairly mild assumption since the probability of two samples having the same angle is zero.

First note that since all the positive examples belong to the positive circular sector, the smallest positive circular sector returned by the algorithm, $h$, is a subset of the positive circular sector $f^{-1}(1)$. Indeed, $f^{-1}(1)$ can be partitioned into three circular sectors $S_1$, $S_2$, and $h$ such that $S_1$ and $S_2$ correspond to the only two area where $h(x) \neq f(x)$ for any $x$. Therefore, by the additivity of measures

$$\mathcal{D}(\{x \mid h(x) \neq f(x)\}) = \mathcal{D}(S_1) + \mathcal{D}(S_2) \tag{12}$$

For $\mathcal{D}(S_1) + \mathcal{D}(S_2)$ to be greater than $\epsilon$, we must have $\mathcal{D}(S_1) > \epsilon/2$ or $\mathcal{D}(S_2) > \epsilon/2$. Therefore, by the union bound

$$\Pr\left[\mathcal{D}(\{x \mid h(x) \neq f(x)\}) > \epsilon\right] \leq \Pr\left[\mathcal{D}(S_1) > \epsilon/2\right] + \Pr\left[\mathcal{D}(S_2) > \epsilon/2\right] \tag{13}$$

Let us focus on the first term involving $S_1$. Assume that we have $m$ examples. We know that none of our $m$ examples belong to $S_1$, or the algorithm would have returned a different, larger, hypothesis. The probability that none of our $m$ samples fell in $S_1$ is $(1 - \mathcal{D}(S_1))^m$, so the probability that $\mathcal{D}(S_1)$ be greater than $\epsilon/2$ is at least $(1 - \epsilon/2)^m$. The same argument holds for $S_2$ so

$$\Pr\left[\mathcal{D}(\{x \mid h(x) \neq f(x)\}) > \epsilon\right] \leq 2(1 - \epsilon/2)^m \tag{14}$$

As $m$ increases, $2(1 - \epsilon/2)^m$ decreases, so for any $\delta < 1$, we can choose $m$ such that $2(1 - \epsilon/2)^m$ is smaller than $1 - \delta$. $\qquad\square$

This theorem holds for any distribution $\mathcal{D}$ and it is worth noting that this includes distributions which satisfies $\mathcal{D}(S_1) = 0$. In such a case, the selected hypothesis $h$ cannot get close to $f$ regardless of how many samples we draw, but the algorithm is still correct since $\mathcal{D}(\{x \mid h(x) \neq f(x) \wedge x \in S_1\}) = 0$. This remark will be important later on in our presentation.

## B.2 Fairness

Assume that we have some predicate $P(.,.)$ which is true on $\mathcal{D}$, and $f$.

As an example, for some constant $k$, such a predicate could be defined as

$$\frac{\lambda(f^{-1}(0))}{\lambda(f^{-1}(1))} = k \tag{15}$$

where $\lambda$ is the Lebesgues measure. This predicate simply states that the ratio of the area of the two circular sectors is a constant.

**Pure learning bias.** Assuming that we have a training set $\mathcal{S}$ and $h = A_{ERM}(\mathcal{D}, f)$, the probability that $P(\mathcal{S}, h)$ holds is 0. However, because $A_{ERM}$ is probably approximately correct, we know that we can use a training set large enough that $\frac{\lambda(h^{-1}(0))}{\lambda(h^{-1}(1))}$ tends to $k$.

**Data Generation bias.** If the samples are drawn from a distribution $\mathcal{D}'$ that is different than $\mathcal{D}$.

**Labeling bias.** If the samples are labeled by a function $f'$ which is different than $f$.

Figure 11: Labeling with $f'$. The positive region of $f'$ is green, the negative region of $f'$ is blue or red, with the red region indicating where $f$ and $f'$ are different.

There are other predicates we could be interested in

$$\frac{\mathcal{D}(f^{-1}(0))}{\mathcal{D}(f^{-1}(1))} = k \tag{16}$$

## C  Empirical Risk Minimization

What happens if we use algorithm $A_{ERM}$ on a training set generated with $(\mathcal{D}, f')$? Then, some of the samples which would have been labeled as positive by $f$ are labeled as negative by $f'$. In consequence, with respect to $f'$, $A_{ERM}$ returns a smaller circular sector than with respect to $f$.

Figure 12: $A_{ERM}$ returns a circular sector of location $\hat{\mu}$ and spread $\hat{\rho}$

This has two implications when we assess the error of $A_{ERM}$.

Figure 13: Error of $A_{ERM}$ w.r.t. $f'$

Figure 14: Error of $A_{ERM}$ w.r.t. $f$

1. With respect to $f'$, $A_{ERM}$ is probably approximately correct
2. With respect to $f$, $A_{ERM}$ has an error lower bound of $\mathcal{D}(S_1) + \mathcal{D}(S_2)$

## D  Empirical Risk Minimization with Empirical Fairness Maximization

We now design an algorithm $A_{EFM}$ which is such that for any training set $\mathcal{S}$, we return an hypothesis $h$ for which $P(\mathcal{D}, h)$ holds. Since $P(\mathcal{D}, h)$ must hold, we have

$$\lambda(h^{-1}(1)) = \frac{\lambda(h^{-1}(0))}{k} \tag{17}$$

and we also know

$$\lambda(h^{-1}(0)) + \lambda(h^{-1}(1)) = 2\pi \tag{18}$$

so we conclude

$$\lambda(h^{-1}(1)) = \frac{2\pi}{k+1} \tag{19}$$

and therefore the spread of $h$ must be $360/(k+1)$.

---

**input**         : A set $\mathcal{S}$ of $N$ labeled examples $(r_j\ e^{i\phi_j}, l_j)$ for $j \in [N]$
**precondition** : $\mathcal{S}$ contains at least 3 positive examples with distinct angles
**output**        : A hypothesis $h \in \mathcal{H}$
1  $(e^{i\mu}, .) = A_{ERM}(\mathcal{S})$;
2  **return** $(e^{i\mu}, 360/(k+1))$

---

**Algorithm 2:** $A_{EFM}$: Smallest positive circular sector with fairness maximization

Figure 15: $A_{EFM}$ returns a circular sector of location $\hat{\mu}$ and spread $\hat{\theta}$

First note that $A_{EFM}$ is not probably approximately correct with respect to $f'$. Indeed, since $\lambda(f'^{-1}(1))$ is smaller than $\lambda(f^{-1}(1))$, $\lambda(f'^{-1}(1)) < 360/(k+1)$.

We will now prove that $A_{EFM}$ is probably approximately correct wirth respect to $f$. Intuitively this is because $h$ has the right spread by construction and with enough data, we should be able to have a good estimate of the location of $f$ with high probability.

**Theorem 2** (Algorithm $A_{EFM}$ is probably approximately correct for positive densities). *For any accuracy $\epsilon > 0$ and confidence $0 < \delta < 1$, if $\mathcal{D}$ is psitive then there exists a finite number $m$ such that if we take $m$ independent samples $e_1, ..., e_m$ from $\mathcal{D}$ and let $h = A_{ERM}(\{(e_i, f(e_i))\})$ we have*

$$\Pr\left[\mathcal{D}(\{x \mid h(x) \neq f(x)\}) > \epsilon\right] \leq 1 - \delta$$

*Proof.* Since $f'$ is contained in $f$, the location of $f'$ is in the circular sector of $f$. Therefore, $h$ and $f$ overlap and their union can be partioned in three different sets, $S_1 = h - f$, $f \cap h$, and $S_2 = f - h$. By the additivity of measures we conclude

$$\mathcal{D}(\{x \mid h(x) \neq f(x)\}) = \mathcal{D}(S_1) + \mathcal{D}(S_2) \tag{20}$$

For $\mathcal{D}(S_1) + \mathcal{D}(S_2)$ to be greater than $\epsilon$, we must have $\mathcal{D}(S_1) > \epsilon/2$ or $\mathcal{D}(S_2) > \epsilon/2$. Therefore, by the union bound

$$\Pr\left[\mathcal{D}(\{x \mid h(x) \neq f(x)\}) > \epsilon\right] \leq \Pr\left[\mathcal{D}(S_1) > \epsilon/2\right] + \Pr\left[\mathcal{D}(S_2) > \epsilon/2\right] \tag{21}$$

Without loss of generality, let us focus on sector $S_1$. There are two other sectors that are relevant in analyzing $S_1$. The first one is the sector defined going from the location of $f$ to the location of $h$ in trigonometric order, we will refer to this sector as $S_c$. The second one is the error of $h$ with respect to $f'$, we will refer to this sector as $S_e$. Note that $S_1$, $S_c$, and $S_e$ all have the same area.

If we assume that the density is non-negative, then there exists some positive $\epsilon'$ such that

$$\Pr[\mathcal{D}(S_1) \geq \epsilon/2] \leq \Pr[\mathcal{D}(S_c) \geq \epsilon'] \tag{22}$$

and likewise, there exists some positive $\epsilon''$ such that

$$\Pr[\mathcal{D}(S_c) \geq \epsilon/2] \leq \Pr[\mathcal{D}(S_e) \geq \epsilon''] \tag{23}$$

Finally, we know from theorem 1 that for all $\epsilon$, $\Pr[\mathcal{D}(S_1') > \epsilon] < (1 - \epsilon)^m$. We can conclude that there exists some decreasing function $a$ such that

$$\Pr[\mathcal{D}(S_1) \geq \epsilon/2] \leq a(m) \tag{24}$$

The same arguments holds for $S_2$ and since $a$ is a decreasing function of $m$, for any value $\delta$ we can choose $m$ large enough. □

Figure 16: Error of $A_{EFM}$ w.r.t. $f'$

Figure 17: Error of $A_{EFM}$ w.r.t. $f$

1. With respect to $f$, $A_{EFM}$ is probably approximately correct for non-negative densities
2. With respect to $f'$, $A_{EFM}$ has an error lower bound of $\mathcal{D}(S_1) + \mathcal{D}(S_2)$