[Reviews · NeurIPS 2019]

Reviewer 1



-The proposed objective function sounds like a reasonable idea except that the unlabel data is assumed fair. This seems like a critical limitation of the proposition. - It is not obvious how to gather unlabeled features having the law of D^’, which is essential to use the semi-supervised learning algorithm. In practice, testing the fairness condition for real data can be more helpful. - All experiments were conducted using synthetic data, and there is no analytics for real datasets. - Used examples are not good and somewhat confusing. For example, in NLP, the difference in modern past and present data can be caused by time-varying distribution. However, the authors regarded this selection bias only. - To strengthen the paper, many measures of AI fairness should be considered. -This paper does not provide downloadable software Minor points: - Page 3, Background, line 5: Maybe replacing ‘protected dimensions’ by ‘protected attributes’ is better. - Page 4, formula (1): p(y|x) -> p(y|x,w) - Page 6, Data generation, line 13: Where -> where - There are no captions to the Figures 1 and 2, and the example to illustrate a hypothetical situation is very difficult to follow. (-) Downloadable source code is provided but a link to a downloadable version of the dataset or simulation environment is not provided.

Reviewer 2



The premise of the paper is that many other papers in the ML fairness literature assume that there is an inevitable trade-off between fairness and accuracy, often without adequate justification for this assumption, and that many papers either assume that the data itself is unbiased or at least do not explicitly their assumptions about the types of bias in the data. I am not fully convinced that the paper's characterization of previous work is accurate. In my view, most fairness papers typically work in one of the following two regimes: 1. The setting in which the learner has access to fair, correct ground truth. In this case, there is clearly no trade-off between fairness and accuracy; if the training and test data are fair, a perfectly accurate classifier would also be perfectly fair. (Some papers that study this setting even state this observation explicitly.) In this setting, most papers (that I am aware of) study whether fairness can also be ensured for imperfect classifiers, since training a perfect classifier is typically not feasible in practice. For instance, they would compare some error metrics (e.g. false positive/negative rates) for different subpopulations. 2. The setting in which the learner does NOT have access to fair, correct ground truth; i.e., the available data is biased, perhaps reflecting historical discrimination or contemporary biases of the people or processes that generated the data. In this setting, there would still be no trade-off between fairness and *true* accuracy, but since the data is assumed *not* to reflect the ideal fair truth, there is a trade-off between fairness and accuracy *as measured on incorrect and biased data.* I agree with the authors that there are many ML papers that don't state their assumptions as clearly as they should. However, many prominent papers in the field do. Also, this paper seems to attribute a confusion of the two regimes I described above to previous work, by saying that "most theoretical work in fair machine learning considers the regime in which neither [the data distribution nor the labeling function] is biased, and most empirical work [...] draws conclusions assuming the regime in which neither is biased" (i.e., the first regime), but also claiming that the most papers assume a trade-off between fairness and ostensible accuracy (i.e., the second regime). Papers that I am familiar with don't confuse the two regimes; they clearly operate in only one of the two, even when they don't make this as clear and explicit as they ideally should. Also, several papers do state these assumptions explicitly, including stating that they assume label bias and the reduction in accuracy for their proposed fair ML methods is due to the fact that they are evaluated on imperfect, biased data. Since the authors talk about the "prevailing wisdom," "most work," etc. in their paper, it's difficult to evaluate whether it's their assessment of the fairness literature or mine that is more accurate; however, I am concerned that they might be attacking a straw man. One of the main contributions of this paper is a set of experiments using synthetic data simulating the second setting above, including synthetically generated bias, so that trained models can be evaluated both on the biased data (that a realistic learner would have access to) and the unbiased ground truth (that in practice is not available to the learner). The analysis is more detailed than in previous work, but the idea is less novel than the paper makes it appear. In the related work section, right after quoting Fish et al., the authors state: "in contrast, we are able to successfully control for label bias because we simulate data ..." Presumably, the "in contrast" refers at least in part to a purported contrast with Fish et al., which is incorrect. In fact, one of the contributions of that cited paper is a method for evaluating fair ML methods on simulated label bias, which is conceptually very similar to what this paper does. The authors don't seem to acknowledge this closely related prior work anywhere in their paper. The other main contribution of the paper is a new, semi-supervised method for fair learning (where the fairness notion used is statistical parity). This fair learning method is essentially ERM with an added unfairness penalty term; from this perspective, the method is very similar to many previously proposed variants of the same idea of using a fairness regularizer. However, the way the authors use unlabeled data for this goal is novel and interesting. The authors provide experimental results with an interesting detailed analysis. However, unfortunately they only compare their own baselines and one previous method from the literature. Given the large number of proposed fair ML methods, many of which operate in a similar setting as this paper and optimize for similar goals, it is disappointing that the authors don't give a more comprehensive comparison with a larger number of previous methods. Finally, a note about language. I appreciate that the authors attempted to write their paper in a more eloquent language than the typical bland and unnatural English that most research papers use. However, given the broad audience of these papers from many languages and cultures, there is sometimes a good reason for using the lowest common denominator of English. Some of the flowery phrases of this paper ("torrents of ink have been spilled") seem unnecessary. Also, the use foreign phrases when there is a simple English equivalent is not effective writing, and serves no practical purpose other than displaying the authors' erudition. For instance, I don't see any reason to use "cum grano salis" instead of the well-known English equivalent of "with a grain of salt." The same holds for "dernier cri," etc. Overall, the results are interesting and the paper is well written, but the main ideas of the paper (adding an unfairness penalty to ERM, evaluating models on synthetic bias) are not entirely novel, and the paper's engagement with previous work needs improvement. Some more minor, detailed comments: - The name of "empirical fairness maximization" for the proposed method seems a little misleading; in reality, the method still minimized empirical risk, with an unfairness penalty term.  "Fair ERM" would be a more precise name, and is a term that previous papers used for similar methods.- In the data generation, it is unclear why the authors decided to generate such a small data set. They explain why they wanted n >> k but an n still small enough to make learning challenging; however, they do not explain why they couldn't have had more features and a larger sample while preserving the relationship between n and k. The paper's data set size and dimensionality are very small compared to typical modern applications.- In the footnote on page 6, the authors mention that instead of accuracy, they use F1 to measure the quality of their models due to the class imbalance in the data. However, since the authors control the data generation process, it's unclear why they couldn't have produced data with balanced labels. - AFTER AUTHOR RESPONSE: I thank the authors for their thoughtful response to my concerns. I acknowledge that I might have underestimated the confusion about label bias in the community; clarifying the assumptions about bias in fair ML is a valuable contribution. I updated my overall score for the submission.

Reviewer 3



This paper provides a very compelling reframing of the common understanding that fairness and accuracy must be traded-off in any fairness-aware algorithm. Instead, this paper convincingly argues that fairness and accuracy can be aligned depending on assumptions about skew in labels and training data selection. These assumptions are clearly articulated, and experiments are provided for varying amounts of skew. This first contribution - the reframing of the fairness / accuracy tradeoff - is a critical one for the fair-ML literature and has the potential to be highly impactful within the subfield, especially given the strength of the articulation of the problem in the first few sections of this paper. Specifically, the paper argues that label bias and selection bias in the training set can lead to unfairness in a way that simultaneously decreases accuracy. The second main contribution of the paper is a fair semi-supervised learning approach. This approach builds on the discussion of the tradeoff by assuming that the labels should be distributed according to a “we’re all equal” assumption. Given this assumption, the SSL technique is a fairly straightforward addition of a term to the loss function. Experiments are then included based on synthetic data generated via a Bayesian network (see Supp Info) to allow for control of the amount of label and selection bias in the resulting data. The resulting experiments show that in the case of label bias, the traditionally accepted fairness / accuracy tradeoff does not apply, and increasing fairness increases accuracy under the stated assumptions. The semi-supervised approach does a good job of achieving both fairness and accuracy.

[Author Response · NeurIPS 2019]

We thank all the reviewers for their helpful comments and suggestions.

**Reviewer 1**   *RE: needing fair unlabeled data.* Just to clarify, when you say we need fair unlabeled data, do you mean
unlabeled data ($D$) without selection bias? In that case, point taken, but we would argue that in some cases label bias a
worse problem, and that to some extent, a good data scientist can help mitigate some of the known biases in the data $D'$
to make it look more like what $D$. Moreover, if you could identify a subset of the data that is less biased, then maybe
you could use this subset to improve the more biased subset through semi-supervision.

*RE: synthetic data.* Synthetic data is a powerful tool that is frequently employed in ML. For some scientific questions,
like ours, it is the only empirical way to study them. If NeurIPS requires experiments on real world data then we would
have to find another venue. Unfortunately, real-world data does not have the observable quantities we need to measure
the true accuracy which we require to investigate the relationship between fairness and accuracy. We could follow R3's
suggestion, but in the end, this still requires synthetic modifications to the data. We had initially considered a similar
idea, but opted to discuss the perspective learning theory provides instead.

**Reviewer 2**   First, thank you very much for the thorough and thoughtful review, especially about the related work. It
is very helpful and constructive.

*RE: related work*   We actually agree with your characterization of the literature and see it as complementary to ours.
Perhaps our language was sometimes too strong in our attempt to highlight what we perceive as a problem. E.g., the use
of "most" instead of "many" may be severe. At the very least, "many" seems appropriate, and as you say, many fairness
papers — including some highly cited papers by highly respected authors — fail to clearly state the assumptions. Take
for example, Zemel et al *Learning Fair Representations*, one of the most highly cited papers in the area. The authors
defer to two previous papers for accuracy reporting (Kamishima et al 2011 and Kamiran&Calders 2009), both of
which discuss accuracy and fairness, but neither of which acknowledge the problem of label bias in that discussion.
Conclusions are then drawn, discussed, disseminated (and repeated) without the assumptions needed to interpret them.[1]

It's true that some papers do mention label bias, and we directly quote the relevant passage from such a paper in the
related work. Though, our perception is that more papers (than not) fail to mention label bias (and this might be where
our characterizations of the literature differ). We believe that label bias is omitted frequently enough that it affects
people's thinking on matters of fairness. For example, sometimes when presenting the semi-supervision idea to friends
in the field for the first time, it is initially dismissed because of the alleged tradeoff. Once assumptions are accurately
communicated and agreed upon, everything becomes copacetic again and the idea is accepted as realistic. To speculate:
part of the problem is that in machine learning, we're so used to our gold standard labels being the indisputable truth
(modulo Cohen's kappa), that it's easy to overlook label bias in fair ML, for which that's no longer true.

Thanks for pointing out our error about the Fish et al paper, which indeed does use simulated data to get access
to the unbiased labels. They are exploring different questions than us, and so they don't end up investigating the
accuracy fairness tradeoff, which we see as one of the main contributions of our paper and a significant distinguishing
characteristic of our work.

*RE: contribution*   We agree that adding yet another term to an objective function is not usually particularly novel, but
in this case it first requires getting over the intellectual hump formed by the fairness accuracy trade-off. Maybe this
is obvious, but the idea was initially met with criticism because of this purported obstacle. The synthetic bias is a
mechanism that we use to investigate the fairness-accuracy tradeoff and we see the investigation itself as a key part of
the contribution.

*RE: other comments.*   (a) Agree that fair ERM might be a better term (b) Data set size was chosen to accommodate
dimensionality while being small for computational expedience (we repeat each experiment 10x so there is actually
10 times more data in total), but in retrospect we could've made it larger (c) it's non-trivial (but possible via more
sophisticated rejection sampling methods) to adjust for perfect class balance while turning the various other experimental
knobs (like protected/unprotected ratio) since there are some dependencies between the experimental parameters;
fortunately, the imbalance was not severe. (d) We will look at the language and try to simplify it for a broader audience.

**Reviewer 3**   This review very accurately identifies our specific contributions, and their relative importance to each
other. We initially planned on doing something similar to your suggestion (3) in which we synthetically modified the
COMPAS data, but decided to use the space for the learning theory example instead. Maybe a more compact way of
conveying the same data would free up some room as you suggested in (2). With regard to (1), we cited that work as an
example of in-processing and used the same objective function as our semi-supervised method, but on the testing set
instead. Perhaps we should've cited a classification paper for in-processing too.

## Footnotes

[1]To be clear, we have great respect for these works and these researchers, and are simply using them to illustrate the point.


[Meta-Review · NeurIPS 2019]

This paper received considerable discussion. Ultimately the reviewers reached consensus that this paper should be accepted -despite- finding the paper lacking in both the strength of the technical contribution and the experimental validation. Reviewers found that the ideas introduced in this work were more interesting and potentially significant than most papers. One reviewer noted that the paper, despite all of its shortcomings, has already changed the way s/he thinks about some elements of algorithmic fairness. *IMPORTANT* Again, I iterate that, while there is strong enthusiasm for the ideas, reviewers remain dissatisfied with the execution. The authors would be doing a disservice to their own work by not making every effort to improve the manuscript to strengthen the technical and experimental contributions prior to publication. This is a case where reviewers are willing to take a chance on a paper in hopes that the authors take initiative to improve their work in the camera-ready.